# Editing Large Language Models: Problems, Methods, and Opportunities

**Yunzhi Yao♣♠,∗, Peng Wang♣♠,∗, Bozhong Tian♣♠, Siyuan Cheng♣♠, Zhoubo Li♣♠,**
**Shumin Deng♡, Huajun Chen♣♠◇, Ningyu Zhang♣♠,†**

♣ Zhejiang University ♠ Zhejiang University - Ant Group Joint Laboratory of Knowledge Graph
◇Donghai Laboratory ♡ National University of Singapore
{yyztodd,peng2001,tbozhong,sycheng,zhoubo.li}@zju.edu.cn
{huajunsir,zhangningyu}@zju.edu.cn,shumin@nus.edu.sg

## Abstract

Despite the ability to train capable LLMs, the methodology for maintaining their relevancy and rectifying errors remains elusive. To this end, the past few years have witnessed a surge in techniques for editing LLMs, the objective of which is to **efficiently** alter the behavior of LLMs within a specific domain without negatively impacting performance across other inputs. This paper embarks on a deep exploration of the problems, methods, and opportunities related to model editing for LLMs. In particular, we provide an exhaustive overview of the task definition and challenges associated with model editing, along with an in-depth empirical analysis of the most progressive methods currently at our disposal. We also build a new benchmark dataset to facilitate a more robust evaluation and pinpoint enduring issues intrinsic to existing techniques. Our objective is to provide valuable insights into the effectiveness and feasibility of each editing technique, thereby assisting the community in making informed decisions on the selection of the most appropriate method for a specific task or context[1].

## 1 Introduction

Large language models (LLMs) have demonstrated a remarkable capacity for understanding and generating human-like text (Brown et al., 2020; OpenAI, 2023; Anil et al., 2023; Touvron et al., 2023; Qiao et al., 2022; Zhao et al., 2023). Despite the proficiency in training LLMs, the strategies for ensuring their relevance and fixing their bugs remain unclear. Ideally, as the world's state evolves, we aim to update LLMs in a way that sidesteps the computational burden associated with training a wholly new model. As shown in Figure 1, to address this issue, the concept of **model editing** has been proposed

---

∗Equal contribution.
†Corresponding author.
[1]Code and datasets are available at https://github.com/zjunlp/EasyEdit.

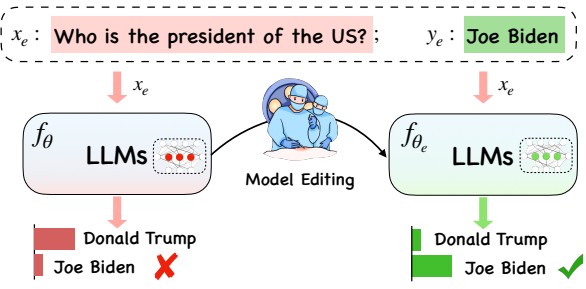

Figure 1: Model editing to fix and update LLMs.

(Sinitsin et al., 2020; De Cao et al., 2021), enabling data-efficient alterations to the behavior of models, specifically within a designated realm of interest, while ensuring no adverse impact on other inputs.

Currently, numerous works on model editing for LLMs (De Cao et al., 2021; Meng et al., 2022, 2023; Sinitsin et al., 2020; Huang et al., 2023) have made strides in various editing tasks and settings. As illustrated in Figure 2, these works manipulate the model's output for specific cases by either integrating an auxiliary network with the original unchanged model or altering the model parameters responsible for the undesirable output. Despite the wide range of model editing techniques present in the literature, a comprehensive comparative analysis, assessing these methods in uniform experimental conditions, is notably lacking. This absence of direct comparison impairs our ability to discern the relative merits and demerits of each approach, consequently hindering our comprehension of their adaptability across different problem domains.

To confront this issue, the present study endeavors to establish a standard problem definition accompanied by a meticulous appraisal of these methods (§2, §3). We conduct experiments under regulated conditions, fostering an impartial comparison of their respective strengths and weaknesses (§4). We initially use two popular model editing datasets, ZsRE (Levy et al., 2017) and COUN-TERFACT (Meng et al., 2022), and two structurally

different language models, T5 (Raffel et al., 2020a) (encoder-decoder) and GPT-J (Wang and Komatsuzaki, 2021a) (decoder only), as our base models. We also evaluate the performance of larger models, OPT-13B (Zhang et al., 2022a) and GPT-NEOX-20B (Black et al., 2022). Beyond basic edit settings, we assess performance for **batch** and **sequential** editing. While we observe that current methods have demonstrated considerable capacity in factual model editing tasks, we reconsider the current evaluation and create a more encompassing evaluation dataset(§5): **portability** (robust generalization capabilities), **locality** (side effect), and **efficiency** (time and memory usage). We find current model editing methods are somewhat limited on these levels, thereby constraining their practical application, and deserve more research in the future. Through systematic evaluation, we aim to provide valuable insights on each model editing technique's effectiveness, aiding researchers in choosing the appropriate method for specific tasks.

## 2 Problems Definition

Model editing, as elucidated by Mitchell et al. (2022b), aims to adjust an initial base model's ($f_\theta$, $\theta$ signifies the model's parameters) behavior on the particular edit descriptor $(x_e, y_e)$ **efficiently** without influencing the model behavior on other samples. The ultimate goal is to create an edited model, denoted $f_{\theta_e}$. Specifically, the basic model $f_\theta$ is represented by a function $f : \mathbb{X} \mapsto \mathbb{Y}$ that associates an input $x$ with its corresponding prediction $y$. Given an edit descriptor comprising the edit input $x_e$ and edit label $y_e$ such that $f_\theta(x_e) \neq y_e$, the post-edit model $f_{\theta_e}$ is designed to produce the expected output, where $f_{\theta_e}(x_e) = y_e$.

The model editing process generally impacts the predictions for a broad set of inputs that are closely associated with the edit example. This collection of inputs is called the **editing scope**. A successful edit should adjust the model's behavior for examples within the editing scope while leaving its performance for out-of-scope examples unaltered:

$$f_{\theta_e}(x) = \begin{cases} y_e & \text{if } x \in I(x_e, y_e) \\ f_\theta(x) & \text{if } x \in O(x_e, y_e) \end{cases} \quad (1)$$

The *in-scope* $I(x_e, y_e)$ usually encompasses $x_e$ along with its equivalence neighborhood $N(x_e, y_e)$, which includes related input/output pairs. In contrast, the *out-of-scope* $O(x_e, y_e)$ consists of inputs that are unrelated to the edit example. The post-edit

model $f_e$ should satisfy the following three properties: **reliability**, **generalization**, and **locality**.

**Reliability**  Previous works (Huang et al., 2023; De Cao et al., 2021; Meng et al., 2022) define a reliable edit when the post-edit model $f_{\theta_e}$ gives the target answer for the case $(x_e, y_e)$ to be edited. The reliability is measured as the average accuracy on the edit case:

$$\mathbb{E}_{x'_e, y'_e \sim \{(x_e, y_e)\}} \mathbb{1} \left\{ \text{argmax}_y f_{\theta_e}\left(y \mid x'_e\right) = y'_e \right\} \quad (2)$$

**Generalization**  The post-edit model $f_{\theta_e}$ should also edit the equivalent neighbour $N(x_e, y_e)$ (e.g. rephrased sentences). It is evaluated by the average accuracy of the model $f_{\theta_e}$ on examples drawn uniformly from the equivalence neighborhood:

$$\mathbb{E}_{x'_e, y'_e \sim N(x_e, y_e)} \mathbb{1} \left\{ \text{argmax}_y f_{\theta_e}\left(y \mid x'_e\right) = y'_e \right\} \quad (3)$$

**Locality**  also noted as **Specificity** in some work. Editing should be implemented locally, which means the post-edit model $f_{\theta_e}$ should not change the output of the irrelevant examples in the out-of-scope $O(x_e, y_e)$. Hence, the locality is evaluated by the rate at which the post-edit model $f_{\theta_e}$'s predictions are unchanged as the pre-edit $f_\theta$ model:

$$\mathbb{E}_{x'_e, y'_e \sim O(x_e, y_e)} \mathbb{1} \left\{ f_{\theta_e}\left(y \mid x'_e\right) = f_\theta\left(y \mid x'_e\right) \right\} \quad (4)$$

## 3 Current Methods

Current model editing methods for LLMs can be categorized into two main paradigms as shown in Figure 2: modifying the model's parameters or preserving the model's parameters. More comparisons can be seen in Table 6.

### 3.1 Methods for Preserving LLMs' Parameters

**Memory-based Model**  This kind of method stores all edit examples explicitly in memory and employs a retriever to extract the most relevant edit facts for each new input to guide the model to generate the edited fact. SERAC (Mitchell et al., 2022b) presents an approach that adopts a distinct *counterfactual model* while leaving the original model unchanged. Specifically, it employs a *scope classifier* to compute the likelihood of new input falling within the purview of stored edit examples. If the input matches any cached edit in memory, the counterfactual model's prediction is based on the input and the most probable edit. Otherwise, if

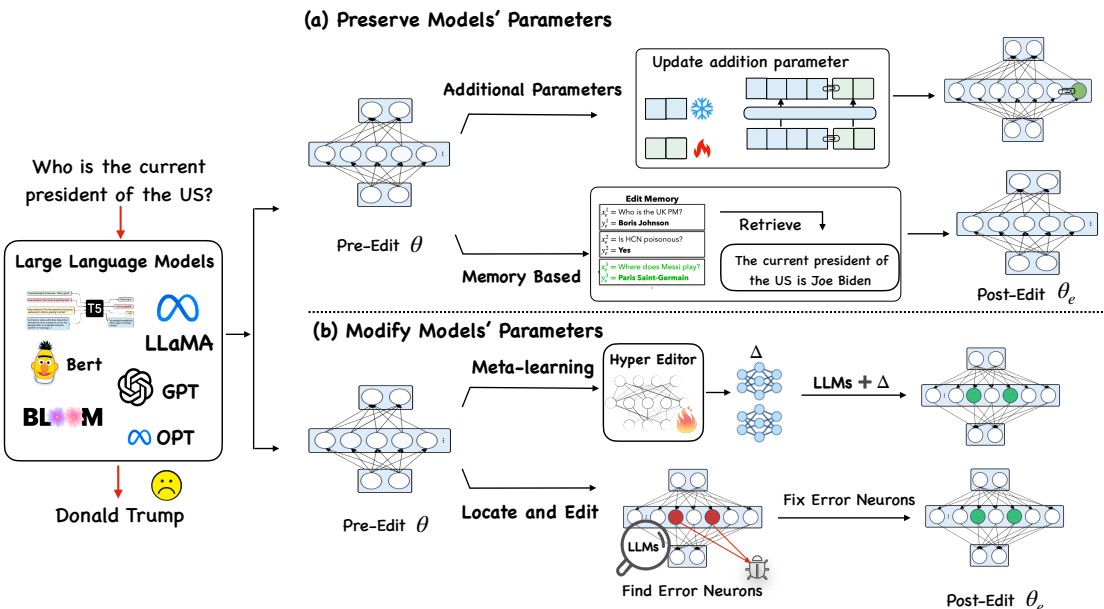

Figure 2: An overview of two paradigms of model editing for LLMs.

the input is out-of-scope for all edits, the original model's prediction is given. Additionally, recent research demonstrates that LLMs possess robust capabilities for *in-context learning*. Instead of resorting to an extra model trained with new facts, the model itself can generate outputs corresponding to the provided knowledge given a refined knowledge context as a prompt. This kind of method edits the language model by prompting the model with the edited fact and retrieved edit demonstrations from the edit memory and includes the following work: MemPrompt (Madaan et al., 2022),IKE (Zheng et al., 2023) and MeLLo (Zhong et al., 2023).

**Additional Parameters** This paradigm introduces extra trainable parameters within the language models. These parameters are trained on a modified knowledge dataset while the original model parameters remain static. T-Patcher (Huang et al., 2023) integrates one neuron(patch) for one mistake in the last layer of the Feed-Forward Network (FFN) of the model, which takes effect only when encountering its corresponding mistake. CaliNET (Dong et al., 2022) incorporates several neurons for multiple edit cases. Differently, GRACE (Hartvigsen et al., 2022) maintains a discrete codebook as an Adapter, adding and updating elements over time to edit a model's predictions.

### 3.2 Methods for Modifying LLMs' Parameters

This paradigm would update part of the parameter $\theta$, it applies an update $\Delta$ matrix to edit the model.

**Locate-Then-Edit** This paradigm initially identifies parameters corresponding to specific knowledge and modifies them through direct updates to the target parameters. The Knowledge Neuron (KN) method (Dai et al., 2022) introduces a *knowledge attribution* technique to pinpoint the "knowledge neuron" (a key-value pair in the FFN matrix) that embodies the knowledge and then updates these neurons. ROME (Meng et al., 2022) applies causal mediation analysis to locate the editing area. Instead of modifying the knowledge neurons in the FFN, ROME alters the entire matrix. ROME views model editing as the least squares with a linear equality constraint and uses the Lagrange multiplier to solve it. However, KN and ROME can only edit one factual association at a time. To this end, MEMIT (Meng et al., 2023) expands on the setup of ROME, realizing the situation of synchronous editing for multiple cases. Based on MEMIT, PMET (Li et al., 2023a) involves the attention value to get a better performance.

**Meta-learning** Meta-learning methods employ a hyper network to learn the necessary $\Delta$ for editing the LLMs. Knowledge Editor (KE) (De Cao et al., 2021) leverages a hypernetwork (specifically, a bidirectional-LSTM) to predict the weight update for each data point, thereby enabling the constrained optimization of editing target knowledge without disrupting others. However, this approach falls short when it comes to editing LLMs. To overcome this limitation, Model Editor Networks

| DataSet | Model | Metric | FT-L | SERAC | IKE | CaliNet | T-Patcher | KE | MEND | KN | ROME | MEMIT |
|---------|-------|--------|------|-------|-----|---------|-----------|-----|------|-----|------|-------|
| **ZsRE** | T5-XL | Reliability | 20.71 | **99.80** | 67.00 | 5.17 | 30.52 | 3.00 | 78.80 | 22.51 | - | - |
| | | Generalization | 19.68 | **99.66** | 67.11 | 4.81 | 30.53 | 5.40 | 89.80 | 22.70 | - | - |
| | | Locality | 89.01 | 98.13 | 63.60 | 72.47 | 77.10 | 96.43 | **98.45** | 16.43 | - | - |
| | GPT-J | Reliability | 54.70 | 90.16 | **99.96** | 22.72 | 97.12 | 6.60 | 98.15 | 11.34 | 99.18 | 99.23 |
| | | Generalization | 49.20 | 89.96 | **99.87** | 0.12 | 94.95 | 7.80 | 97.66 | 9.40 | 94.90 | 87.16 |
| | | Locality | 37.24 | 99.90 | 59.21 | 12.03 | 96.24 | 94.18 | 97.39 | 90.03 | 99.19 | 99.62 |
| **COUNTERFACT** | T5-XL | Reliability | 33.57 | **99.89** | 97.77 | 7.76 | 80.26 | 1.00 | 81.40 | 47.86 | - | - |
| | | Generalization | 23.54 | **98.71** | 82.99 | 7.57 | 21.73 | 1.40 | 93.40 | 46.78 | - | - |
| | | Locality | 72.72 | **99.93** | 37.76 | 27.75 | 85.09 | 96.28 | 91.58 | 57.10 | - | - |
| | GPT-J | Reliability | 99.90 | 99.78 | 99.61 | 43.58 | **100.00** | 13.40 | 73.80 | 1.66 | 99.80 | 99.90 |
| | | Generalization | 97.53 | **99.41** | 72.67 | 0.66 | 83.98 | 11.00 | 74.20 | 1.38 | 86.63 | 73.13 |
| | | Locality | 1.02 | **98.89** | 35.57 | 2.69 | 8.37 | 94.38 | 93.75 | 58.28 | 93.61 | 97.17 |

Table 1: Results of existing methods on three metrics of the dataset. The settings for these models and datasets are the same with Meng et al. (2022). '-' refers to the results that the methods empirically fail to edit LLMs.

with Gradient Decomposition (MEND) (Mitchell et al., 2022a) learns to transform the gradient of fine-tuned language models by employing a low-rank decomposition of gradients, which can be applied to LLMs with better performance.

## 4 Preliminary Experiments

Considering the abundance of studies and datasets centered on factual knowledge, we use it as our primary comparison foundation. Our initial controlled experiments, conducted using two prominent factual knowledge datasets (Table 1), facilitate a direct comparison of methods, highlighting their unique strengths and limitations (Wang et al., 2023b).

### 4.1 Experiment Setting

We use two prominent model editing datasets: ZsRE and COUNTERFACT, with their details available in Appendix B. Previous studies typically used smaller language models (<1B) and demonstrated the effectiveness of current editing methods on smaller models like BERT (Devlin et al., 2019). However, whether these methods work for larger models is still unexplored. Hence, considering the editing task and future developments, we focus on generation-based models and choose larger ones: T5-XL (3B) and GPT-J (6B), representing both encoder-decoder and decoder-only structures.

We've selected influential works from each method type. Alongside existing model editing techniques, we additionally examined the results of fine-tuning, an elementary approach for model updating. To avoid the computational cost of retraining all layers, we employed methodology proposed by Meng et al. (2022), fine-tuning layers identified by ROME and we denoted it as FT-L. This strategy ensures a fair comparison with other direct editing

methods, bolstering our analysis's validity. More details can be found in Appendix A.

### 4.2 Experiment Results

**Basic Model** Table 1 reveals SERAC and ROME's superior performance on the ZsRE and COUNTERFACT datasets, with SERAC exceeding 90% on several metrics. While MEMIT lacks its generalization, it excels in reliability and locality. KE, CaliNET, and KN perform poorly, with acceptable performance in smaller models, but mediocrity in larger ones. MEND performs well on the two datasets, achieving over 80% in the results on T5, although not as impressive as ROME and SERAC. The performance of the T-Patcher model fluctuates across different model architectures and sizes. For instance, it underperforms on T5-XL for the ZsRE dataset, while it performs perfectly on GPT-J. In the case of the COUNTERFACT dataset, T-Patcher achieves satisfactory reliability and locality on T5 but lacks generalization. Conversely, on GPT-J, the model excels in reliability and generalization but underperforms in the locality. This instability can be attributed to the model architecture since T-Patcher adds a neuron to the final decoder layer for T5; however, the encoder may still retain the original knowledge. FT-L performs less impressively than ROME on PLMs, even when modifying the same position. It shows underwhelming performance on the ZsRE dataset but equals ROME in reliability and generalization with the COUNTERFACT dataset on GPT-J. Yet, its low locality score suggests potential impacts on unrelated knowledge areas. IKE demonstrates good reliability but struggles with locality, as prepended prompts might affect unrelated inputs. Its generalization capability could also improve. The in-context learning

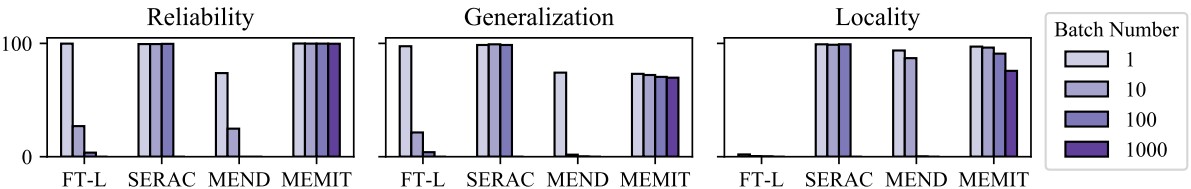

Figure 3: **Batch Editing** performance against batch number. We test batch numbers in [1,10,100,1000] for MEMIT. Due to the huge memory usage for FT, SERAC and MEND, we didn't test batch 1000 for these methods.

method may struggle with context mediation failure (Hernandez et al., 2023), as pre-trained language models may not consistently generate text aligned with the prompt.

**Model Scaling** We conduct experiments with larger models, testing IKE, ROME, and MEMIT on OPT-13B and GPT-NEOX-20B due to computational constraints. The results (Table 2) surprisingly show ROME and MEMIT performing well on the GPT-NEOX-20B model but failing on OPT-13B. This is due to both methods relying on a *matrix inversion* operation. However, in the OPT-13B model, the matrix is not *invertible*. We even empirically find that approximating the solution with least squares yields unsatisfactory results. We think this is the limitation of ROME and MEMIT as they are based on the strong assumption that matrices are non-degenerate and may not be applied to different models. MEMIT performs worse due to its reliance on multi-layer matrix computations, and its reliability and generalization declined more than ROME's for larger models. IKE's performance is affected by the in-context learning ability of the model itself. The results of OPT are even worse than the results of GPT-J, which may be attributed to OPT's own in-context learning ability. Additionally, as the model size increases, its performance in both generalization and locality diminishes.

**Batch Editing** We conduct further batch editing analysis, given that many studies often limit updates to a few dozen facts or focus only on single-edit cases. However, it's often necessary to modify the model with multiple knowledge pieces simultaneously. We focused on batch-editing-supportive methods (FT, SERAC, MEND, and MEMIT) and displayed their performance in Figure 3. Notably, MEMIT supports massive knowledge editing for LLMs, allowing hundreds or even thousands of simultaneous edits with minimal time and memory costs. Its performance across reliability and generalization remains robust up to 1000 edits, but lo-

| Method | ZSRE | | | COUNTERFACT | | |
|--------|------|------|------|------|------|------|
| | Reliability | Generalization | Locality | Reliability | Generalization | Locality |
| *OPT-13B* | | | | | | |
| ROME | 22.23 | 6.08 | **99.74** | 36.85 | 2.86 | **95.46** |
| MEMIT | 7.95 | 2.87 | 92.61 | 4.95 | 0.36 | 93.28 |
| IKE | 69.97 | **69.93** | 64.83 | **49.71** | **34.98** | 53.08 |
| *GPT-NEOX-20B* | | | | | | |
| ROME | 99.34 | 95.49 | **99.79** | **99.80** | **85.45** | 94.54 |
| MEMIT | 77.30 | 71.44 | 99.67 | 87.22 | 70.26 | **96.48** |
| IKE | **100.00** | **99.95** | 59.69 | 98.64 | 67.67 | 43.03 |

Table 2: Current methods' results of current datasets on **OPT-13B** and **GPT-NEOX-20B**.

cality decreases at this level. While FT-L, SERAC, and MEND also support batch editing, they require significant memory for handling more cases, exceeding our current capabilities. Thus, we limited tests to 100 edits. SERAC can conduct batch edits perfectly up to 100 edits. MEND and FT-L performance in batch edits is not as strong, with the model's performance rapidly declining as the number of edits increases.

**Sequential Editing** Note that the default evaluation procedure is to update a single model knowledge, evaluate the new model, and then roll back the update before repeating the process for each test point. In practical scenarios, models should retain previous changes while conducting new edits. Thus, the ability to carry out successive edits is a vital feature for model editing (Huang et al., 2023). We evaluate approaches with strong single-edit performance for sequential editing and report the results in Figure 4. Methods that freeze the model's parameters, like SERAC and T-Patcher, generally show stable performance in sequential editing. However, those altering the model's parameters struggle. ROME performs well up to $n = 10$, then degrades at $n = 100$. MEMIT's performance also decreases over 100 edits, but less drastically than ROME. Similarly, MEND performs well at $n = 1$ but significantly declines at $n = 10$. As the editing process continues, these models increasingly deviate from their original state, resulting in suboptimal performance.

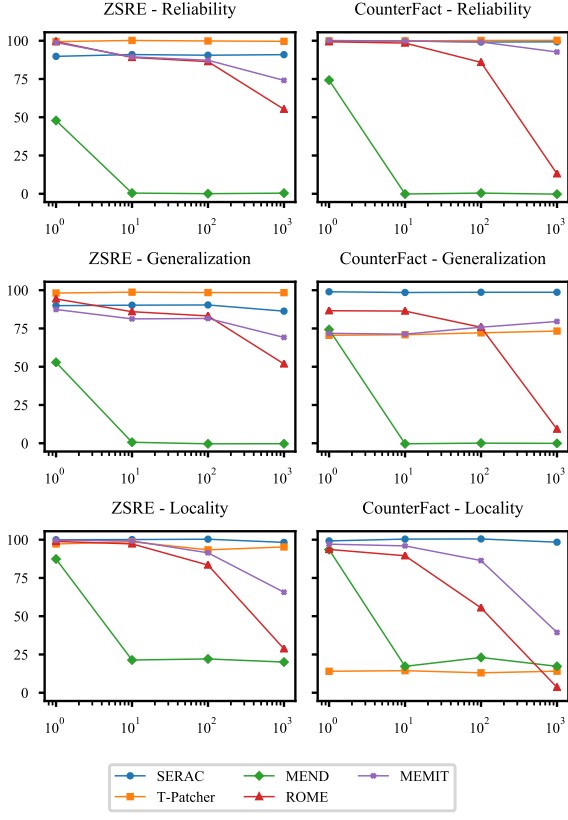

Figure 4: **Sequential Editing** performance against data stream size (log-scale).

| Method | Subject-Replace | Reverse-Relation | One-hop |
|---|---|---|---|
| *GPT-J-6B* | | | |
| FT-L | 72.96 | 8.05 | 1.34 |
| SERAC | 17.79 | 1.30 | 5.53 |
| T-Patcher | **96.65** | 33.62 | 3.10 |
| MEND | 42.45 | 0.00 | 11.34 |
| ROME | 37.42 | 46.42 | 50.91 |
| MEMIT | 27.73 | 47.67 | 52.74 |
| IKE | 88.77 | **92.96** | **55.38** |
| *GPT-NEOX-20B* | | | |
| ROME | 44.57 | 48.99 | 51.03 |
| MEMIT | 30.98 | 49.19 | 49.58 |
| IKE | **85.54** | **96.46** | **58.97** |

Table 3: Portability results on various model editing methods. The example for each assessment type can be found in Table 7 at Appendix B.

## 5 Comprehensive Study

Considering the above points, we contend that previous evaluation metrics may not fully assess model editing capabilities. Therefore, we propose more comprehensive evaluations regarding **portability**, **locality**, and **efficiency**.

### 5.1 Portability - Robust Generalization

Several studies evaluate generalization using samples generated through back translation (De Cao et al., 2021). However, these paraphrased sentences often involve only minor wording changes and don't reflect substantial factual modifications. As stated in Jacques Thibodeau (2022), it's crucial to verify if these methods can handle the implications of an edit for realistic applications. As a result, we introduce a new evaluation metric called **Portability** to gauge the effectiveness of model editing in transferring knowledge to related content, termed robust generalization. Hence we consider three aspects: (1) **Subject Replace**: As most rephrased sentences keep subject descriptions but rephrase the relation more, we test generalization by replac-

ing the subject in the question with an alias or synonym. This tests whether the model can generalize the edited attribute to other descriptions of the same subject. (2) **Reversed Relation**: When the target of a subject and relation is edited, the attribute of the target entity also changes. We test the model's ability to handle this by filtering for suitable relations such as one-to-one and asking it the reverse question to check if the target entity is also updated. (3) **One-hop**: Modified knowledge should be usable by the edited language model for downstream tasks. For example, if we change the answer to the question "What university did Watts Humphrey attend?" from "Trinity College" to "University of Michigan", the model should then answer "Ann Arbor in Michigan State" instead of "Dublin in Ireland" when asked, "Which city did Watts Humphrey live in during his university studies?" We thus construct a reasoning dataset to evaluate the post-edit models' abilities to use the edited knowledge.

We incorporate a new part, $P(x_e, y_e)$, into the existing dataset ZsRE, and **Portability** is calculated as the average accuracy of the edited model ($f_{\theta_e}$) when applied to reasoning examples in $P(x_e, y_e)$:

$$\mathbb{E}_{x'_e, y'_e \sim P(x_e, y_e)} \mathbb{1} \left\{ \operatorname{argmax}_y f_{\theta_e} \left( y \mid x'_e \right) = y'_e \right\} \tag{5}$$

**Dataset Construction** As to the one-hop dataset, in the original edit, we alter the answer from $o$ to $o^*$ for a subject $s$. We then prompt the model to generate a linked triple $(o^*, r^*, o'^*)$. Subsequently, GPT-4 creates a question and answer based on this triple and $s$. Notably, if the model can answer

this new question, it would imply that it has pre-existing knowledge of the triple $(o^*, r^*, o'^*)$. We filter out unknown triples by asking the model to predict $o'^*$ from $o^*$ and $r^*$. If successful, it's inferred the model has prior knowledge. Finally, **Human evaluators** verify the triple's accuracy and the question's fluency. Additional details, such as the demonstrations we used and other parts of dataset construction, can be found in the Appendix B.

**Results** We conduct experiments based on the newly proposed evaluation metric and datasets, presenting the results in Table 3. As demonstrated by the Table, the performance of current model editing methods regarding portability is somewhat suboptimal. SERAC, despite showing impeccable results on previous metrics, scores less than 20% accuracy across all three portability aspects. The bottleneck of SERAC lies in the accuracy of the classifier and the capabilities of the additional model. As to the *subject replace* scenario, including SERAC, MEND, ROME, and MEMIT, can only adapt to a specific subject entity expression but cannot generalize to the concept of the subject entity. However, FT-L, IKE, and T-patcher demonstrate great performance when facing the substituted subject. Regarding the *reversed relation*, our results indicate that current editing methods mainly edit one-direction relations, with IKE as the notable exception, achieving over 90% on both GPT-J and GPT-NEOX-20B. Other methods alter the subject entities' attributes while leaving the object entity unaffected. In the *one-hop* reasoning setting, most of the editing methods struggle to transfer the altered knowledge to related facts. Unexpectedly, ROME, MEMIT, and IKE exhibit relatively commendable performance on portability (exceeding 50%). They are capable of not only editing the original cases but also modifying facts correlated with them in some respect. To summarize, IKE exhibits relatively good performance across the three scenarios in our evaluations. However, it is clear that current model editing techniques continue to face challenges in managing the ramifications of an edit - that is, ensuring that changes to knowledge are coherently and consistently reflected in related contexts. This area, indeed, calls for further investigation and innovation in future research.

## 5.2 Locality - Side Effect of Model Editing

In the preceding section, COUNTERFACT and ZsRE evaluate model editing's locality from differ-

| Method | Other-Attribution | Distract-Neighbor | Other-Task |
|---|---|---|---|
| FT-L | 12.88 | 9.48 | 49.56 |
| MEND | 73.50 | 32.96 | 48.86 |
| SERAC | **99.50** | 39.18 | 74.84 |
| T-Patcher | 91.51 | 17.56 | 75.03 |
| ROME | 78.94 | 50.35 | 52.12 |
| MEMIT | 86.78 | 60.47 | 74.62 |
| IKE | 84.13 | **66.04** | **75.33** |

Table 4: Locality results on various model editing methods for GPT-J. Examples of each type can be seen in Tabel 9 at Appendix B.

ent perspectives. COUNTERFACT employs triples from the same distribution as the target knowledge, while ZsRE utilizes questions from the distinct Natural Questions dataset. Notably, some methods, such as T-Patcher, exhibit differing performances on these two datasets. This highlights that the impact of model editing on the language model is multifaceted, necessitating a thorough and comprehensive evaluation to fully appreciate its effects. To thoroughly examine the potential side effects of model editing, we propose evaluations at three different levels: (1) **Other Relations**: Although Meng et al. (2022) introduced the concept of *essence*, they did not explicitly evaluate it. We argue that other attributes of the subject that have been updated should remain unchanged after editing. (2) **Distract Neighbourhood**: Hoelscher-Obermaier et al. (2023a) find that if we concatenate the edited cases before other unrelated input, the model tends to be swayed by the edited fact and continue to produce results aligned with the edited cases. (3) **Other Tasks**: Building upon Skill Neuron's assertion (Wang et al., 2022) that feed-forward networks in large language models (LLMs) possess task-specific knowledge capabilities, we introduce a new challenge to assess whether model editing might negatively impact performance on other tasks. Construction of the dataset details can be found in Appendix B.3.

**Results** Table 4 presents our results. Notably, current editing methods excel in the *other attributions* aspect, indicating that they only modify the target characteristic without affecting other attributes. However, they generally perform poorly in *Distract-Neighbor* settings, as reflected in the performance drop compared to the results in Table 1. An exception is IKE, whose performance remains relatively stable due to the fact that it inher-

| Editor | COUNTERFACT | ZsRE |
|---|---|---|
| FT-L | 35.94s | 58.86s |
| SERAC | 5.31s | 6.51s |
| CaliNet | 1.88s | 1.93s |
| T-Patcher | 1864.74s | 1825.15s |
| KE | 2.20s | 2.21s |
| MEND | **0.51s** | **0.52s** |
| KN | 225.43s | 173.57s |
| ROME | 147.2s | 183.0s |
| MEMIT | 143.2s | 145.6s |

Table 5: **Wall clock time** for each edit method conducting 10 edits on GPT-J using one 2×V100 (32G). The calculation of this time involves measuring the duration from providing the edited case to obtaining the post-edited model.

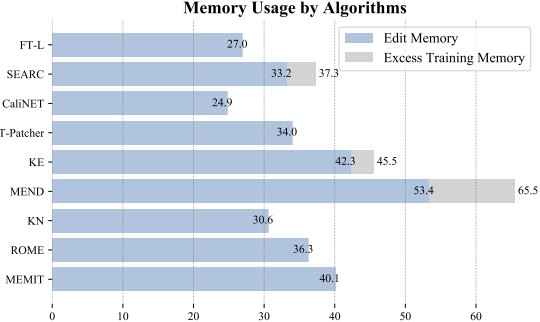

Figure 5: **GPU VRAM consumption during training and editing** for different model editing methods.

ently requires the edited fact to be concatenated before the input. As for the commonsense reasoning tasks, parameter-preserving methods largely maintain their performance on other tasks. Conversely, methods that alter parameters tend to negatively influence performance, with the exception of MEMIT. Despite changing parameters, MEMIT maintains strong performance in commonsense tasks, demonstrating its commendable locality.

### 5.3 Efficiency

Model editing should minimize the **time** and **memory** required for conducting edits without compromising the model's performance.

**Time Analysis** Table 5 illustrates the time required for different model editing techniques from providing the edited case to obtaining the post-edited model. We observe that once the hypernetwork is trained, KE and MEND perform the editing process at a considerably fast pace. Likewise, SERAC can also quickly edit knowledge, completing the process in about 5 seconds, given a trained classifier and counterfact model. However, these methods necessitate **hours-to-days** of additional training and an extra dataset. In our experiments, training MEND on the ZsRE dataset took over 7 hours, and training SERAC required over 36 hours on 3× V100. On the other hand, ROME and MEMIT necessitate a pre-computation of the covariance statistics for the Wikitext. However, this computation is time-consuming and can potentially take **hours-to-days** to complete. In comparison, other methods such as KN, CaliNET, and T-Patcher may be faster since they do not require any pre-computation or pre-training. However, KN and CaliNET's performance on larger models is

unsatisfactory, and T-Patcher is the slowest due to the need for individual neuron training for each corresponding mistake. Considering the time aspect, there is a need for a model editing method that is more time-friendly.

**Memory Analysis** Figure 5 exhibits the memory VRAM usage for each model editing method. From this figure, we observe that the majority of the methods consume a similar amount of memory, with the exception of MEND, which requires more than 60GB for training. Methods that introduce extra training, such as MEND and SERAC lead to additional computational overhead, hence the significant increase in memory consumption.

## 6 Relationship with Relevant Works

### 6.1 Knowledge in LLMs

Several model editing approaches aim to discern how knowledge stored in PLMs precisely and directly alters the model's parameters. There is existing work that examines the principles that govern how PLMs store knowledge (Geva et al., 2021, 2022; Haviv et al., 2023; Hao et al., 2021; Hernandez et al., 2023; Yao et al., 2023; Cao et al., 2023; Lamparth and Reuel, 2023; Cheng et al., 2023; Li et al., 2023b; Chen et al., 2023; Ju and Zhang, 2023), which contribute to the model editing process. Moreover, some model editing techniques bear resemblance to knowledge augmentation (Zhang et al., 2019; Lewis et al., 2020; Zhang et al., 2022b; Yasunaga et al., 2021; Yao et al., 2022; Pan et al., 2023) approaches, as updating the model's knowledge can also be considered as instilling knowledge into the model.

## 6.2 Lifelong Learning and Unlearning

Model editing, encompassing lifelong learning and unlearning, allows adaptive addition, modification, and removal of knowledge. Continual learning (Biesialska et al., 2020), which improves model adaptability across tasks and domains, has shown effectiveness in model editing in PLMs (Zhu et al., 2020). Moreover, it's vital for models to forget sensitive knowledge, aligning with machine unlearning concepts (Hase et al., 2023; Wu et al., 2022; Tarun et al., 2021; Gandikota et al., 2023).

## 6.3 Security and Privacy for LLMs

Past studies (Carlini et al., 2020; Shen et al., 2023) show that LLMs can produce unreliable or personal samples from certain prompts. The task of erasing potentially harmful and private information stored in large language models (LLMs) is vital to enhance the privacy and security of LLM-based applications (Sun et al., 2023). Model editing, which can suppress harmful language generation (Geva et al., 2022; Hu et al., 2023), could help address these concerns.

## 7 Conclusion

We systematically analyze methods for editing large language models (LLMs). We aim to help researchers better understand existing editing techniques by examining their features, strengths, and limitations. Our analysis shows much room for improvement, especially in terms of portability, locality, and efficiency. Improved LLM editing could help better align them with the changing needs and values of users. We hope that our work spurs progress on open issues and further research.

## Acknowledgment

We would like to express gratitude to the anonymous reviewers for their kind comments. This work was supported by the National Natural Science Foundation of China (No.62206246), Zhejiang Provincial Natural Science Foundation of China (No. LGG22F030011), Ningbo Natural Science Foundation (2021J190), Yongjiang Talent Introduction Programme (2021A-156-G), CCF-Tencent Rhino-Bird Open Research Fund, and Information Technology Center and State Key Lab of CAD&CG, Zhejiang University.

## Limitations

There remain several aspects of model editing that are not covered in this paper.

**Model Scale & Architecture** Due to computational resource constraints, we have only calculated the results for models up to 20B in size here. Meanwhile, many model editing methods treat the FFN of the model as key-value pairs. Whether these methods are effective for models with different architectures, such as Llama, remains to be explored.

**Editing Scope** Notably, the application of model editing goes beyond mere factual contexts, underscoring its vast potential. Elements such as personality, emotions, opinions, and beliefs also fall within the scope of model editing. While these aspects have been somewhat explored, they remain relatively uncharted territories and thus are not detailed in this paper. Furthermore, multilingual editing (Xu et al., 2022; Wang et al., 2023a; Wu et al., 2023) represents an essential research direction that warrants future attention and exploration. There are also some editing works that can deal with computer vision tasks such as ENN (Sinitsin et al., 2020) and Ilharco et al. (2023).

**Editing Setting** In our paper, the comprehensive study 5 mainly evaluated the method's performance on one edit. During the time of our work, Zhong et al. (2023) proposed a multi-hop reasoning setting that explored current editing methods' generalization performance for multiple edits simultaneously. We leave this multiple-edit evaluation for the future. Besides, this work focused on changing the model's result to reflect specific facts. Cohen et al. (2023) propose a benchmark for knowledge injection and knowledge update. However, erasing the knowledge or information stored in LLMs (Belrose et al., 2023; Geva et al., 2022; Ishibashi and Shimodaira, 2023) is also an important direction for investigating.

**Editing Black-Box LLMs** Meanwhile, models like ChatGPT and GPT-4 exhibit remarkable performance on a wide range of natural language tasks but are only accessible through APIs. This raises an important question: How can we edit these "black-box" models that also tend to produce undesirable outputs during downstream usage? Presently, there are some works that utilize in-context learning (Onoe et al., 2023) and prompt-based methods (Murty et al., 2022) to modify these models.

They precede each example with a textual prompt that specifies the adaptation target, which shows promise as a technique for model editing.

## Ethic Consideration

Model editing pertains to the methods used to alter the behavior of pre-trained models. However, it's essential to bear in mind that ill-intentioned model editing could lead the model to generate harmful or inappropriate outputs. Therefore, ensuring safe and responsible practices in model editing is of paramount importance. The application of such techniques should be guided by ethical considerations, and there should be safeguards to prevent misuse and the production of harmful results. **All our data has been carefully checked by humans, and any malicious editing or offensive content has been removed**.

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

## A Implementing Details

Since the ZsRE dataset adopts the NQ dataset to evaluate the locality, here, we use a T5-XL model (Raffel et al., 2020b) finetuned on the NQ dataset as the baseline model. As to the GPT-J (Wang and Komatsuzaki, 2021b), we use the original pre-trained version to test the locality's zero-shot results. As several original implementations do not support both architectures, we have re-implemented them to accommodate both models. We re-implemented some original implementations to support both models. However, our empirical findings suggest that ROME and MEMIT are only suitable for decoder-only models like GPT-J, so we have not reported results for T5-XL.

**FT** For basic Fine-Tuning (FT), we follow Meng et al. (2023) re-implementation in their study, which uses Adam (Kingma and Ba, 2014) with early stopping to minimize $-log\mathbb{P}_{G'}[o* \mid p]$, changing only $mlp_{proj}$ weights at selected layer 21. For both models, all hyperparameters follow default settings. To ensure fairness in the experiments, we always use the unconstrained fine-tuning approach.

**KE** De Cao et al. (2021) develops an LSTM sequence model, which employs gradient information to predict the rank-1 weight alterations in $G$. Given that the official code doesn't facilitate GPT-J, we resort to using the re-implemented version provided by Mitchell et al. (2022a) in their research. To foster an equitable comparison across both zsRE and COUNTERFACT tasks, we have taken additional steps to train KE-zsRE and KE-CF models. The hyperparameters employed for training have been sourced from the default configurations provided. During testing, KE presents a scaling factor to tweak the norm of the weight update, for which we adhere to the default value of 1.0.

| | | Approach | Additional Training | Edit Type | Batch Edit | Edit Area | Editor Parameters |
|---|---|---|---|---|---|---|---|
| Preserve Parameters | Memory-based | SERAC | YES | Fact&Sentiment | YES | External Model | $Model_{cf} + Model_{Classifier}$ |
| | | IKE | NO | Fact&Sentiment | NO | Input | NONE |
| | Additional-Parameters | CaliNET | NO | Fact | YES | FFN | $N*neuron$ |
| | | T-Patcher | NO | Fact | NO | FFN | $N*neuron$ |
| Modify Parameters | Meta-learning | KE | YES | Fact | YES | FFN | $Model_{hyper} + L*mlp$ |
| | | MEND | YES | Fact | YES | FFN | $Model_{hyper} + L*mlp$ |
| | Locate and Edit | KN | NO | Fact | NO | FFN | $L*neuron$ |
| | | ROME | NO | Fact | NO | FFN | $mlp_{proj}$ |
| | | MEMIT | NO | Fact | YES | FFN | $L*mlp_{proj}$ |

Table 6: Comparisons between several existing model editing approaches. "Additional Training" refers to whether the methods need training before conducting specific edits. "Edit Type" refers to the format the method can edit. "Batch Edit" refers to editing multiple target knowledge simultaneously. "Editor Area" refers to the specific region of the LLMs that the methods aim to modify. FFN demonstrates the feed-forward module. "Editor Parameters" refers to the parameters that need to be updated for editing. $L$ denotes the number of layers to update. $mlp$ is FFN and $mlp_{proj}$ is the second linear layer in FFN. $neurons$ denotes the key-value pair in FFN. $N$ represents the quantity of $neuron$ to be updated within a single layer.

**CaliNET** Dong et al. (2022) enriches the FFN by incorporating extra parameters aimed at knowledge editing, comprised of a number of calibration memory slots. In order to adapt CaliNET to the task at hand, we retain the same architecture as used in the Feed-Forward Network (FFN), albeit with a reduced intermediate dimension denoted as $d$. This adaptation allows us to effectively apply CaliNET while managing the model's complexity and computational requirements. Regarding hyperparameters, we implement adjustments to the FFN within the final two layers of GPT-J, while all other configurations remain consistent with the default settings.

**MEND** Mitchell et al. (2022a) develop an efficient method for locally editing language models using just a single input-output pair. Essentially, MEND employs a technique to manipulate the gradient of fine-tuned language models which leverages a low-rank decomposition of gradients. The hyperparameters follow default settings, with the exception of several experiments conducted on GPT-J. Specifically, we adjust the optimizer from Adam to AdamW.

**SERAC** Mitchell et al. (2022b) presents a method for model editing, named MEME (Memory-Based Model Editing), which stores edits in an explicit memory and learns to reason over them to adjust the base model's predictions as needed. The system uses an explicit cache of user-provided edit descriptors (arbitrary utterances for language models), alongside a small auxiliary *scope classifier* and *counterfactual model*. The scope classifier determines if the input falls within the scope of any cached items, and if so, the counterfactual model uses the input and the most relevant edit example to make a prediction.

In alignment with the original paper, we use publicly available Huggingface implementations and checkpoints for all experiments. For the SERAC *scope classifier* model, we adopt `distilbert-base-cased` (Sanh et al., 2019) across all models and experimental settings. For the counterfactual model, we employ `T5-small` (Raffel et al., 2020b) for the T5-XL implementation and `architext/gptj-162M` (available at here[2]) for the GPT-J implementation. All hyperparameters for training and test-time inference are derived from default configurations.

Similar to T-Patcher, in auto-regressive model (like GPT-J) training, we only consider loss at the output positions.

**KN** For Knowledge Neuron (Dai et al., 2022), we follow Meng et al. (2023) re-implementation in their study. The method begins by identifying neurons that are closely associated with knowledge expression. This selection is made through gradient-based attributions, which effectively highlight the neurons that have a strong influence on the model's output. After these critical neurons are identified, the method modifies the projection layer of the feed-forward network (denoted as $mlp_{proj}^{(l)}$) specifically at the rows corresponding to the selected neurons. This modification involves adding scaled embedding vectors to the current values, effectively adjusting the model's behavior in a targeted manner. Specifically, they amplify knowledge neurons

---

[2] https://huggingface.co/architext/gptj-162M

by doubling their activations. Similar to FT, all hyperparameters are adopted from default configurations(See code[3])

**T-Patcher**  The method proposed by Huang et al. (2023) offers a way to alter the behavior of transformer-based models with minimal changes. Specifically, it adds and trains a small number of neurons in the last Feed-Forward Network (FFN) layer. This approach effectively provides a means for fine-tuning model behavior with less computational demand than comprehensive retraining. It freezes all original parameters and adds one neuron (patch) to the last FFN layer for one mistake. And they train the patch to take effect only when encountering its corresponding mistake. For T5-XL implementation, all hyperparameters follow the same default settings as Bart-base[4].

Furthermore, in the auto-regressive model (like GPT-J), the model may make multiple mistakes in one example. Therefore, for an example where the model makes n mistakes, we only consider errors generated by the model at the **output positions**. Following the settings of the original paper, we add up to 5 patches for one edit example. Formally, for an edit example $(x_e, y_e)$ in auto-regressive model, the actual input is given by $\hat{x}_e = x_e + y_e$ and the patched model's output is $p_e$, $l_e$ is defined as:

$$l_e = -\sum_{i=1}^{N} \hat{x}_i \log(p_i) \cdot \mathbf{1}_{(i \geq len(x_e))} \qquad (6)$$

**ROME**  ROME, as proposed by Meng et al. (2022), conceptualizes the MLP module as a straightforward key-value store. For instance, if the key represents a subject and the value encapsulates knowledge about that subject, then the MLP can reestablish the association by retrieving the value that corresponds to the key. In order to add a new key-value pair, ROME applies a rank-one modification to the weights of the MLP, effectively "writing in" the new information directly. This method allows for more precise and direct modification of the model's knowledge. We directly apply the code and MLP weight provided by the original paper [5] and keep the default setting for hyper-parameters.

**MEMIT**  MEMIT (Meng et al., 2023) builds upon ROME to insert many memories by modifying the MLP weights of a range of critical layers.

We test the ability of MEMIT using their code [6] and all hyperparameters follow the same default settings. For GPT-J, we choose R = 3, 4, 5, 6, 7, 8, and covariance statistics are collected using 100,000 samples of Wikitext. For GPT-NEOX-20B, we select R = 6, 7, 8, 9, 10, and covariance statistics are collected from over 50,000 samples of Wikitext.

**IKE**  IKE (Zheng et al., 2023) defines three types of demonstration formatting templates including (i)copy, (ii)update, (iii)retain, which guide LMs to edit knowledge facts by in-context learning (ICL). As there are no parameter modifications, IKE is applicable to any existing LLMs.

In alignment with the original paper, we choose k-NN examples from the training corpus(10000 size). The demonstrations are encoded by all-MiniLM-L6-v2. Following the default setting, we set k to 32(See code[7]).

## B  Dataset Details

### B.1  Basic DataSet

**ZsRE**  (Levy et al., 2017) is a Question Answering (QA) dataset using question rephrasings generated by back-translation as the equivalence neighborhood. COUNTERFACT (Meng et al., 2022) is a more challenging dataset that accounts for counterfacts that start with low scores in comparison to correct facts. It constructs out-of-scope data by substituting the subject entity for a proximate subject entity sharing a predicate. This alteration enables us to differentiate between superficial wording changes and more significant modifications that correspond to a meaningful shift in a fact. We follow previous data split (De Cao et al., 2021; Meng et al., 2022; Mitchell et al., 2022a) to evaluate all the models on the test set. For models requiring training, we utilize the training set. Following prior work (Mitchell et al., 2022a,b), we use the Natural Questions (NQ; Kwiatkowski et al. (2019)) as out-of-scope data to evaluate locality.

### B.2  Dataset Construction for Portability Evaluation

#### B.2.1  One hop

The construction can be seen in Figure 6. To ensure that the original model($f_\theta$) has seen the triple $(s, r, o)$ during the pre-training process, we employ

---

[3]https://github.com/EleutherAI/knowledge-neurons
[4]https://github.com/ZeroYuHuang/Transformer-Patcher
[5]https://rome.baulab.info/

[6]https://memit.baulab.info/
[7]https://github.com/PKUnlp-icler/IKE

| Type | Edit Descriptor | Portability Question |
|---|---|---|
| **Subject Replace** | In what living being can *PRDM16* be found? When was *Liu Song dynasty* abolished? *Table tennis* was formulated in? | In what living being can *PR domain containing 16* be found? When was the end of *the Former Song* dynasty? *ping pang*, that originated in ? |
| **Reversed Relation** | What is Wenxiu's spouse's name? | Who is the wife/husband of Wenxi Emperor? |
| **One-hop Reason** | What company made Volvo B12M? | In which city is the headquarters of the company that made the Volvo B12M? |

Table 7: Example of portability dataset.

link prediction to predict $o$ given $(s, r, ?)$. If the tail entity is present in the Top-10 logits, we consider the model to have prior knowledge of this triple. In other words, if the model has sufficient portability, it can correctly answer new questions based on the subject and the triplet.

We select data points to measure the performance of the model's portability. The symbolic representation of the portability dataset is as follows:

$$D_{port} = \{\text{GPT4}\,(s, r, o) \mid o \in \text{Top-10}(f_\theta\,(s, r, ?))\}$$

To guide GPT-4 in producing the desired question and answer, we employ a few-shot manual demonstration as a prompt (See Table 10). In addition, we intersect the data filtered by T5-XL and the data filtered by GPT-J to obtain the final portability dataset. The GPTJ model achieves a link prediction score of ZSRE: **72.99** and COUNTERFACT: **69.78**, while the T5 model achieves a link prediction score of ZSRE: **83.90** and COUNTERFACT: **84.81**. It ensures that the models possess prior knowledge about this triple.

As a result, we select some data instances from the **ZsRE** and the **COUNTERFACT** dataset. The description of the data is shown in Table 8.

|  | Subject Replace | Inverse Relation | One hop |
|---|---|---|---|
| **ZsRE** | 293 | 385 | 1,037 |
| **COUNTERFACT** | 213 | - | 1,031 |

Table 8: Statistics of portability dataset.

### B.2.2 Subject Replace

We replace the question's subject with an alias or synonym to test generalization on other descriptions of the subject. We used two approaches to construct this dataset. 1. For subjects that could be found in Wikidata, we replaced the original subject with the alias from Wikidata (Field Name: `Also known as`). 2. For subjects that could not be found

in Wikidata, we used GPT-4 to generate synonyms for the original subject. This process ensures that the evaluation accurately reflects the model's capability to handle various subject representations, contributing to a more comprehensive understanding of its performance.

### B.2.3 Reversed Relation

In an editing instance, the attributes of the target entity can also change. For instance, in the edited instance: "Who is the father of Nebaioth? Ishmael → Babur." When answering the question "Who is the son of Babur?" it should be answered based on the new fact after editing, which is **Nebaioth**. Certain types of relations may not be as effective for evaluation. Let's consider a hypothetical scenario where we change the location of the Eiffel Tower to Rome - proposing a valid reversed question in such a context would be challenging. Consequently, we carefully handpicked all relations in the ZsRE dataset that could be reversed, such as one-to-one relations, and selected related questions through keywords (such as `spouse`, `wife`, `mother`, `father`, `brother`, `sister`) screening. This allows us to maintain the integrity and relevance of the evaluation process, thus ensuring more reliable results. To guide GPT-4 in producing the desired reversed question, we employ a few-shot manual demonstration as a prompt (See Table 11).

### B.3 Dataset Construction for Locality Evaluation

### B.3.1 Other Attribution

Other attributes of the subject updated should remain the same before editing. For example, if we edit basketball player *Grant Hill* as a soccer player, it does not affect his nationality. Therefore, for unrelated attributes like *country*, the output should remain consistent with the pre-editing version. We modified the **COUNTERFACT** dataset by using the Wikidata API to traverse all relationships of a subject and randomly select an unrelated relation-

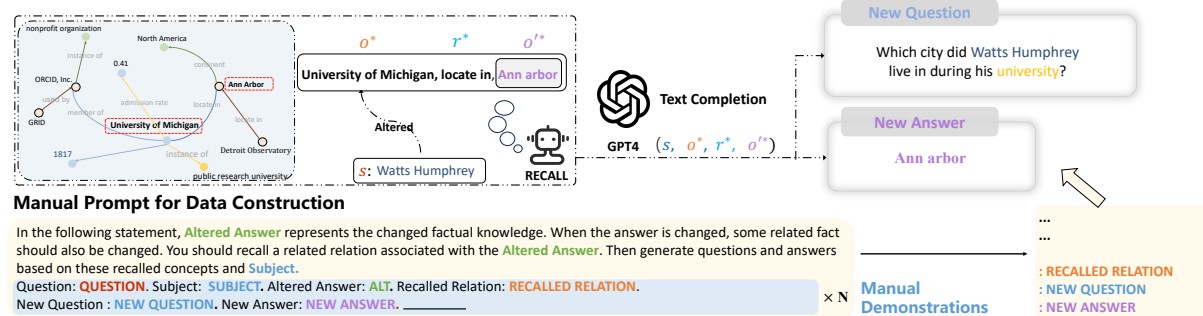

Figure 6: Dataset construction procedure to generate portability part (Q,A) with GPT4.

| Type | Edit Descriptor | Locality Question |
|---|---|---|
| **Other Attribution** | *Grant Hill* is a professional _ | Which country does *Grant Hill* represent in sport? (relation: *country*) |
| | The language of *La Dispute* was _ | What genre does *La Dispute* belong to? (relation: *genre*) |
| | *Gleb Kotelnikov* is a native speaker of _ | What is the gender of *Gleb Kotelnikov*? (relation: *sex or gender*) |
| **Distract Neighbor** | *Windows 98* was a product of _ | *Windows 98* was a product of IBM. Windows Media Center, developed by _ |
| | The language of *Goodfellas* is _ | The language of *Goodfellas* is Tamil. The language of Titanic is _ |

Table 9: Example of locality dataset.

ship and tail entity as a data sample. We provide $(s, r_{other})$ to GPT-4 to generate a question, and the answer to this question corresponds to the respective tail entity. As a result, we modify **804** data instances from the **COUNTERFACT** dataset.

### B.3.2 Distract Neighbor

Following Hoelscher-Obermaier et al. (2023b), we modify the neighborhood prompt in **COUNTER-FACT** dataset by prepending the model edit. For example(See Table 9), if the original prompt is "Windows 98 was a product of _" the modified prompt would be "Windows 98 was a product of IBM. Windows Media Center, developed by _". It measures whether the model editing technique has resulted in significant side effects on the model itself due to over-editing. As a result, we select **804** data instances from the **COUNTERFACT** dataset.

### B.3.3 Other Task

We select commonsense tasks here to assess the post-edited model's performance on other tasks. Given a question $q$, multiple-choice commonsense reasoning aims to select the correct answer $a_t \in \mathcal{A}$ provided with an optional context $c$. **Physical Interaction QA** (PIQA ((Bisk et al., 2020)) is a 2-way multiple-choice QA task testing physics reasoning about objects. We evaluate the post-edit model on the PIQA dataset to reflect the impact of different model editing techniques on the performance of other downstream tasks. Specifically, For each model editing technique, we **sequentially** edit

GPT-J with 100 samples in the **COUNTERFACT** dataset. Afterward, we test the performance of the continuously post-edit models on PIQA, using accuracy as the selected metric, which is defined as :

$$acc = \sum_{k=1}^{N} Q(c_k, q_k, a_{k_p})/N \qquad (7)$$

where $a_{k_p}$ is the option with the least perplexity of the post-edit model, $Q(c_k, q_k, a_{k_p})$ is 1 if $a_{k_p} = a_{k_t}$ and 0 otherwise.

**Prompt**

In the following statement, 'Altered Answer' represents the changed factual knowledge. When the answer is changed, some related facts should also be changed. You should recall a related relation associated with the 'Altered Answer'. Then generate questions and answers based on these recalled concepts and 'Subject'.

---

Question: What university did Watts Humphrey attend?
Subject: Watts Humphrey
Altered Answer: University of Michigan
**Recalled Relation: (University of Michigan, locate in, Ann Arbor)**
New Question: Which city did Watts Humphrey live in during his undergraduate studies?
New Answer: Ann Arbor in Michigan State

---

Question: Windows 10, developed by
Subject: Windows 10
Altered Answer: Google
**Recalled Relation: (Sundar Pichai, ceo of, Google)**
New Question: Who is the CEO of the company that develops the Windows 10 operating system?
New Answer: Sundar Pichai

---

Question: In Kotka, the language spoken is?
Subject: Kotka
Altered Answer: French
**Recalled Relation: (French, evolve from, Romance)**
New Question: What language did Kotka's official language evolve from?
New Answer: Romance

---

Question: Armand Trousseau's area of work is?
Subject: Armand Trousseau
Altered Answer: jazz
**Recalled Relation: (Miles Davis, genres, jazz)**
New Question: Armand Trousseau formed a band during college, they are all fans of?
New Answer: Miles Davis

Table 10: Prompt for dataset construction on zsRE & CounterFact portability dataset. Demonstration examples are manually constructed. For each data instance, we provide Question, Subject, and Altered Answer to generate portability data.

| Task | Prompt |
|------|--------|
| **ZSRE** | Please generate the Inverse Question(For example, A and B are in a father-son relationship. In the original question, it says "who is the father of B? Answer is A". You should ask who is the son/daughter of A, so answer is B.) here are some examples: |

Q: Who is Claire Clairmont's sister? A: Marian Clairmont
Inverse Question: Who is Marian Clairmont's sister?

Q: What was the name of the father of Jane Seymour? A: Richard Seymour
Inverse Question: Who is the son/daughter of Richard Seymour?

Q: What is Elizabeth Grey, Countess of Kent's spouse? A: Henry Grey, 1st Duke of Suffolk
Inverse Question: Who was Henry Grey, 1st Duke of Suffolk married to?

Q: Who is listed as Leonor, Princess of Asturias's father? A: Leonor III of Spain
Inverse Question: Who is the son/daughter of Leonor III of Spain?

Table 11: Prompt for inversed relation dataset construction on zsRE, we provide Question and Answer to generate an inversed question.