# OpenReview forum: "Editing Large Language Models: Problems, Methods, and Opportunities"
_EMNLP/2023/Conference — EMNLP 2023 Main_

### Official Review · Reviewer_yYDP · 2023-08-04

**Soundness:** 3

**Excitement:**

3: Ambivalent: It has merits (e.g., it reports state-of-the-art results, the idea is nice), but there are key weaknesses (e.g., it describes incremental work), and it can significantly benefit from another round of revision. However, I won't object to accepting it if my co-reviewers champion it.

**Paper Topic And Main Contributions:**

This paper provides a systematic analysis and comparative evaluation of different techniques for editing large language models (LLMs). This paper addresses the lack of comprehensive comparative analysis and evaluation of different techniques for editing large language models (LLMs). Existing works have not assessed these methods in uniform experimental conditions. Authors propose a new evaluation metric called Portability to gauge the generalization ability of different model editing methods.

The main contributions are:
1. provides an exhaustive overview of current model editing methods, organizing them into categories based on whether they preserve or modify model parameters.
2. identifies a lack of comparative analysis across editing techniques under consistent experimental conditions.
3. proposes more comprehensive benchmark evaluations on portability, locality, and efficiency to enable a more robust assessment.
4. conducts controlled experiments for impartial comparison of the strengths and weaknesses of each method.
5. aims to offer guidance on selecting the most appropriate editing method for a given task or context based on the comparative analysis.

**Reasons To Accept:**

1. The topic is highly relevant and timely given the rapid progress in LLMs recently. Editing their behavior efficiently is an important challenge and meaningful work to generate content which follows commonsense and up-to-date.
2. The manuscript is clearly structured and well-written overall. The abstract and introduction motivate the gap this analysis aims to fill.
3. Comparing editing models under uniform experimental conditions is a worthwhile contribution. The proposed benchmarks enable more robust evaluation.
4. The focus on practical implications for method selection is important for guiding real-world applications.

**Reasons To Reject:**

1. The scope is currently limited to factual knowledge editing. Expanding the discussion to other editable attributes could increase impact.
2. Authors reach conclusion that MEND's performance on GPT-J is less impressive, which may be attributed to the model size. However, no experiment result on OPT-13B and GPT-NEOX-20B with MEND method available. The conclusion is not convincing.
3. Providing some sample results from the comparative experiments could better illustrate the insights gained. Lack of qualitative analysis.
4. Rather than just reporting performance across metrics, including examples of bad cases, analyzing patterns in these failures, and providing intuitive explanations for why they occurred could strengthen the paper. This deeper perspective enables us to really understand the conditions under which certain editing techniques break down.

**Reproducibility:**

4: Could mostly reproduce the results, but there may be some variation because of sample variance or minor variations in their interpretation of the protocol or method.

**Reviewer Confidence:**

2: Willing to defend my evaluation, but it is fairly likely that I missed some details, didn't understand some central points, or can't be sure about the novelty of the work.

---

> ### Author Rebuttal · Authors · 2023-08-28
>
> Thanks for your valuable reviews, we hope the following comments could address your concerns.
>
> > The scope is currently limited to factual knowledge editing. Expanding the discussion to other editable attributes could increase impact.
>
> This is a great comment.
> Model editing is far from factual knowledge editing and there are several works focused on emotions and writings.
> Our paper select factual editing as the starting point to analyze existing work and reflect existing problems.
> But **the issues summarized in our work are not limited to fact**; they apply to other editable attributes as well.
> Facts can serve as a scaffold to evaluate and address these broader challenges.
> Meanwhile, other editable attributes is a developing field that requires future research.
>
> > Authors reach conclusion that MEND's performance on GPT-J is less impressive, which may be attributed to the model size. However, no experiment result on OPT-13B and GPT-NEOX-20B with MEND method available. The conclusion is not convincing.
>
> Thanks for your constructive comments, and we have conducted more experiments.
> **In our experiments and previous works' results (like ROME, MEMIT), MEND's result on GPT-J(6B) is usually worse than small size model like GPT2-XL(1.7B) and T5_XL(2.8B)**.
> However, we recently conducted experiments on Llama2-7B and Llama, and we found that MEND performs well on these models.
> Despite having the same size, the performance of MEND on Llama is worse than that on Llama2.
> The results is here:
> | Model      | Llama2-7B | Llama-7B
> | ----------- | ----------- | -----------
> | Reliability   |   94.24    | 89.35
> | Generalization     |   90.27     | 82.15
> |  Locality  |   97.04     | 96.62
>
> We are also conducting the results of OPT-13B and GPT-NEO-20B, but the models are too large and training MEND's hyper network requires several days and we cannot report this results currently.
> Furthermore, we observed fluctuations in the performance of the meta-learning method during our experiments. To gain a deeper understanding of these fluctuations, we will conduct a more thorough analysis. Thank you for bringing this to our attention, and we appreciate your understanding.
> **We would modify this part to make it correct after the we get the results of OPT-13B and GPT-NEO-20B**.
>
>
> > Providing some sample results from the comparative experiments could better illustrate the insights gained. Lack of qualitative analysis.
> Rather than just reporting performance across metrics, including examples of bad cases, analyzing patterns in these failures, and providing intuitive explanations for why they occurred could strengthen the paper. This deeper perspective enables us to really understand the conditions under which certain editing techniques break down.
>
> This is a great problem.
> In our Analysis part, **we explore in detail how existing methods perform on different types of knowledge**, such as the reversed relation, distract neighbor and so on.
> And from this part, we can found different methods demonstrates different performance on different type of knowledge.
> For example, SERAC can deal with the given knowledge perfectly but capacity of the small counterfactual model limit its performance on the reversed relation.
> Meanwhile, as mentioned by the reviewer yadK, we propose a detailed explanation for the failure of some methods and please check our explanation for reviewer yadK.
>
> Importantly, we conduct a **bad case analysis** here to give a more deeper perspective.
> First, we found that **KN and SERAC** tend to generate the tokens repeatedly like ''9thththththththththth'.
> Also, we can find a similar pattern for each method and we list one case here.
> | Method    | Bad case |  Label | Output
> | ----------- | ----------- |  ----------- | -----------
> | MEND   |   Which was the production company for Peepli Live?    | Peepli Entertainment | \npli Entertainment
> | SERAC     |   Which lady gave birth to Leto? Fausta - Leto's mom's who?     | Fausta | Estusta
> |  ROME  |   Which was the gender of Dena Feingold?    | male | nobody
>
> 1. MEND usually tend to miss the first word and prefer to generate some meaningless token like'\n'. Actually, they seems to learn the new fact but failed to generate it.
> 2. SERAC suffers from the small model. The small model is not able to learn the new fact well and cannot predict the answer correctly.
> 3. ROME usually tend to generate answers that are unrelated with the answer. In our experiment, we found the computation of this v* vector for updating the model's weight may not success in this error case.
>
> We are doing more deeper analysis and we would add it during the revision.

---

### Official Review · Reviewer_5Cwu · 2023-08-05

**Soundness:** 4

**Excitement:**

4: Strong: This paper deepens the understanding of some phenomenon or lowers the barriers to an existing research direction.

**Paper Topic And Main Contributions:**

This paper discusses the problems, methods, and opportunities related to model editing for LLMs. The authors also introduce a new benchmark dataset and 'Portability' to facilitate a more robust evaluation of models.

**Questions For The Authors:**

This domain isn't very familiar to me, so I'd like to ask about how the approaches discussed would benefit expert and non-expert target audiences?

**Reasons To Accept:**

The authors provide a detailed categorization and discussion of model editing techniques. I appreciate the discussion of harmful language generation as well.

**Reasons To Reject:**

The edits discussed are single-edits, so some discussion on how to evaluate multiple-edits would be good to include.

**Reproducibility:**

4: Could mostly reproduce the results, but there may be some variation because of sample variance or minor variations in their interpretation of the protocol or method.

**Reviewer Confidence:**

1: Not my area, or paper was hard for me to understand. My evaluation is just an educated guess.

---

> ### Author Rebuttal · Authors · 2023-08-28
>
> Thanks for your valuable suggestions.
> For your concern about the multi-edits, **we have conducted both batch editing and sequential editing in Section 4.3** and we list the performance of current editing techniques.
>
> Furthermore, in terms of the benefits of our work, model editing is a rapidly developing field, and there will be more opportunities in the future. Even when we train highly capable models, it remains unclear how to keep them up to date or revise them when they make errors. Ideally, when the state of the world changes, we should be able to update models in a way that is computationally less expensive than training a new model. Although there are some existing methods (as discussed in our paper) that can partially address this problem, these methods still have limitations.
>
> **For experts**, our work highlights urgent issues that need to be addressed to facilitate the exploration of knowledge updating in language models. **For non-expert audiences**, our work provides a list of the advantages and disadvantages of current methods, aiding them in selecting the most suitable one for downstream usage.

---

### Official Review · Reviewer_yadK · 2023-08-10

**Typos Grammar Style And Presentation Improvements:** 1. There are certain places where the…
**Soundness:** 4

**Excitement:**

4: Strong: This paper deepens the understanding of some phenomenon or lowers the barriers to an existing research direction.

**Paper Topic And Main Contributions:**

The manuscript describes the model-editing (ME) problem, where we selectively update the model’s parameters such that the rest of the parameters are kept intact. The main objective is to update the knowledge for given facts/information, while the other information is intact. The work has shown insights into problems, methodologies, and future work in the ME domain.

The work highlights the notations and definitions, recent works, an empirical analysis of the methods, and the challenges.  In total, two datasets and models are experimented with ten different editing techniques. An ablation and set of ME techniques have covered both the paradigms of preserving Models’ parameters and Updating Models’ parameters.

**Questions For The Authors:**

Question A: Do the ROME and MEMIT work for the encoder-decoder or maybe the encoder-only model family? Since in Table 1, it is marked with ‘-’.

Question B: Why are scores in Table 1 for KE<->GPT-J very low? Similarly, for the CaliNet, KN as well.

Question C: In line 341, ROME degrades its performance, but what would be the reason for the performance degradation?

Question D: In Figure 3, what is the y-axis? It is not very clear from the figure alone.

**Reasons To Accept:**

The reasons to accept are as follows:
1. The presentation of the problem is well-written & visualized with the notations, current methods, experiment settings, and results.
2. The manuscript covers the essential and necessary problem of editing. The insights are definitely helpful for the NLP community.
3. The set of ME techniques in Table 1 covers standard editing techniques from two different paradigms. It shows an insightful observation over two different datasets and three metrics.
4. In the Appendix, the implementation details can help in reproducibility. Additionally, the wall clock time and GPU consumption details are an add-on to show the analysis.
5. The manuscript is a good combination of review-research papers. The work showcases the previous works, their challenges, and their comparison. It also adds the contributions of subject-replace, reverse-relation, and one-hop.

**Reasons To Reject:**

The reasons for the rejection are as follows:
1. The models representing the encoder-only family are not present. However, the insights available for the other two: encoder-decoder & decoder-only, in the manuscript are enough to show the insights and observations.
2. A comparison between the batch and sequential techniques for all the editing techniques would also be a great insight.
3. Insights are missing and the observations are directly reported. For instance, the reasoning for the performance degradation after editing with 100 instances is not available in the manuscript.

**Reproducibility:**

4: Could mostly reproduce the results, but there may be some variation because of sample variance or minor variations in their interpretation of the protocol or method.

**Reviewer Confidence:**

4: Quite sure. I tried to check the important points carefully. It's unlikely, though conceivable, that I missed something that should affect my ratings.

---

> ### Author Rebuttal · Authors · 2023-08-28
>
> Thanks for recognizing the value of our work. Your comments are highly aligned with our paper, and we hope the following comments could answer your questions.
>
> Comments:
> > Effectiveness of editing techniques in encoder-only architecture models
>
> Some recent work has explored editing in encoder-only architectures, such as RoBERTa and BERT, demonstrating the effectiveness of these techniques. The results of the paper "Editing Factual Knowledge in Language Models" [De Cao 2021] are as follows:
> | Model      | BERT |
> | ----------- | ----------- |
> | Reliability   |   98.80     |
> | Generalization     |   82.69     |
> |  Locality  |   98.14      |
> In our paper, we skip this part for the following reasons:
> 1. The performance of current editing methods is excellent for models with fewer than 1B parameters. Since the topic of this study is **editing techniques on large language models**, we focus on encoder-decoder and decoder-only models with a large number of parameters. However, encoder-only models like BERT and RoBERTa usually have fewer than 1B parameters.
> 2. **Encoder-only models are primarily designed for classification tasks**. Considering the editing task and future developments, we focus on generation-based models.
>
> Comments:
> > Batch and sequential experiments for all the editing techniques
>
> It is worth noting that **not all methods support batch editing**. As indicated in Table 6 in the appendix, the batch editing methods include SERAC, CaliNET, KE, MEND, MEMIT, and FT-L. However, in the main experiment, we observed that KE and CaliNET exhibited poor editing performance on a model with a large number of parameters. Since our goal was to provide guidance on selecting the most suitable editing method for a given situation, we chose the remaining editing methods for evaluation and performed sequential editing accordingly.
>
> Comments:
> > The reasoning for the performance degradation after editing 100 instances
>
> Thanks for the reminder. Due to the limited space on the page, we cannot provide a detailed explanation. We will include the explanations in the Appendix. Here are the reasons for the degradation:
>
> For the method that changes the original model's parameters, each edit makes the updated matrix far from the original matrix.
> Specially,
> - **KE&MEND**: MEND did not place restrictions on weight updates in the model MLP and update the weight based on the editor trained on the original parameters, thus causing serious damage to the model after 10 edits
> - **ROME & MEMIT**: Through the hard/soft equality constraint for weight update, ROME, MEMIT maintained excellent reliability, generalization, Locality, etc., within 100 samples. However, the computation of the new $\hat{W}$ is also based on the covariance  $C$ and $C$ is computed on the original model. After the editing, the $C$ is not suitable for the post-edit model, leading to the degradation after editing 100 instances.This turning point reflects the ability of different editing methods to accommodate editing samples.
>
> Question A
> >  Do the ROME and MEMIT work for the encoder-decoder and encoder-only models?
>
> In our experiments, it was found that ROME&MEMIT could not work on models on the encoder-decoder architecture.
> Here are two reasons:
> 1. Specifically, both methods make use of matrix inversion to solve the least squares constraint. For example, rome:
> $\Lambda=(v_*-Wk_*)/(C^{-1} k_*)^{T} k_*$,
> MEMIT: $\Delta=R K_{1}^{T}(C_{0}+K_{1} K_{1}^{T})^{-1}$,the initial matrix $W_0$, requirements MLP is reversible.
> However, in our experiment, similar with the OPT-13B model, we find that 'wo.weight' in T5 is a singular matrix, which does not meet the condition of matrix inversion.
> To this end, we have made the following two attempts:
> - Find the pseudo-inverse of the matrix, however, because $w_o$ has a higher dimension (greater than 10000), it cannot afford its time cost.
> - Least squares approximation, for example $\ Delta (C_ {0} + K_ {1} K_ {1} ^ {T}) = RK_ {1} ^ {T} $using ` torch. Linalg. LSTSQ ` approaching $(C_ {0} + K_ {1} K_ {1} ^ {T}) $. This makes ROME and MEMIT completely incapable of editing (1-2% reliability).
>
> 2. ROME and MEMIT conduct causal analysis based on the decoder-only model, which uses the unidirectional attention mechanism. The encoder-decoder model and encoder-only model, on the other hand, utilize the bidirectional attention mechanism. This model can predict the masked token based on the given text.
> However, when we changed the representation of the last subject token, we found that the encoder was still able to predict the changed token by the context surrounded, rendering the edit ineffective.
>
> Overall, We assume that ROME and MEMIT is not able to work for encoder-only and encoder-decoder model.
>
> Question B
> >  Why are scores in Table 1 for KE<->GPT-J very low? Similarly, for the CaliNet, KN as well.
>
> The performance of KE on large models is not good.
> It is a common phenomenon in other works [Mitchell 2022, Huang 2023]. The reason can be attributed to the huge parameters for the large language model, which is difficult to fit.
> **KN** first conduct a gradient attribution to locate the model and we find it is difficult to locate for the GPT-J size model.
> **CaliNET** is a previous work of T-Patcher, it only adds a few tokens and do not apply the constraint loss like T-Patcher. We found the added token usually failed to be activated when answering the question, which lead to the edit failure.
>
> Question C
>
> > In line 341, ROME degrades its performance, but what would be the reason for the performance degradation?
>
> We list the reason in the second comment.
>
> Question D
> > In Figure 3, what is the y-axis? It is not very clear from the figure alone.
>
> The y-axis in Figure 3 represents average accuracy, and thank you for pointing this out. We would make it clear.

---

### Meta-Review · Area_Chair_k7v7 · 2023-09-19

**Recommendation:** 4

**Metareview:**

**quality, clarity, originality**

The paper and its  handling of the topic of model editing is high quality. The writing was very clear and provided clarity in a number of aspects of dealing with ways to edit language models and impacts on performance. The reviewers appreciated the depth of the work as well as how thorough the authors were in handling the topic. The questions that touched on encoder only architectures were answered and context provide to other literature.

The originality of the work in creating a uniform field to explore the language editing approaches expands on the novelty and provides insight for readers and field. Benchmarks will prove useful for others to work to emulate or build upon.

**significance**
The topic itself is very important to the NLP community especially as more and more model reuse is accelerating. The work is significant as it provides insight into the model editing area and also provide data benchmarks that can be reused by others. This serves as a great resource for the field.

**notes for authors**

Please do make sure that the salient points such as providing an overview on encoder models as well as descriptions/discussioins of failures on the editing are documented in the paper itself.

---

### Decision · Program_Chairs · 2023-10-07

**Decision:**

Accept-Main

**Comment:**

**quality, clarity, originality**

The paper and its  handling of the topic of model editing is high quality. The writing was very clear and provided clarity in a number of aspects of dealing with ways to edit language models and impacts on performance. The reviewers appreciated the depth of the work as well as how thorough the authors were in handling the topic. The questions that touched on encoder only architectures were answered and context provide to other literature.

The originality of the work in creating a uniform field to explore the language editing approaches expands on the novelty and provides insight for readers and field. Benchmarks will prove useful for others to work to emulate or build upon.

**significance**
The topic itself is very important to the NLP community especially as more and more model reuse is accelerating. The work is significant as it provides insight into the model editing area and also provide data benchmarks that can be reused by others. This serves as a great resource for the field.

**notes for authors**

Please do make sure that the salient points such as providing an overview on encoder models as well as descriptions/discussioins of failures on the editing are documented in the paper itself.